# GAP Safe screening rules for sparse multi-task and multi-class models

**Eugene Ndiaye    Olivier Fercoq    Alexandre Gramfort    Joseph Salmon**
LTCI, CNRS, Télécom ParisTech, Université Paris-Saclay
Paris, 75013, France
`firstname.lastname@telecom-paristech.fr`

## Abstract

High dimensional regression benefits from sparsity promoting regularizations. Screening rules leverage the known sparsity of the solution by ignoring some variables in the optimization, hence speeding up solvers. When the procedure is proven not to discard features wrongly the rules are said to be *safe*. In this paper we derive new safe rules for generalized linear models regularized with $\ell_1$ and $\ell_1/\ell_2$ norms. The rules are based on duality gap computations and spherical safe regions whose diameters converge to zero. This allows to discard safely more variables, in particular for low regularization parameters. The GAP Safe rule can cope with any iterative solver and we illustrate its performance on coordinate descent for multi-task Lasso, binary and multinomial logistic regression, demonstrating significant speed ups on all tested datasets with respect to previous safe rules.

## 1   Introduction

The computational burden of solving high dimensional regularized regression problem has lead to a vast literature in the last couple of decades to accelerate the algorithmic solvers. With the increasing popularity of $\ell_1$-type regularization ranging from the Lasso [18] or group-Lasso [24] to regularized logistic regression and multi-task learning, many algorithmic methods have emerged to solve the associated optimization problems. Although for the simple $\ell_1$ regularized least square a specific algorithm (*e.g.,* the LARS [8]) can be considered, for more general formulations, penalties, and possibly larger dimension, coordinate descent has proved to be a surprisingly efficient strategy [12].

Our main objective in this work is to propose a technique that can speed-up any solver for such learning problems, and that is particularly well suited for coordinate descent method, thanks to active set strategies.

The *safe rules* introduced by [9] for generalized $\ell_1$ regularized problems, is a set of rules that allows to eliminate features whose associated coefficients are proved to be zero at the optimum. Relaxing the safe rule, one can obtain some more speed-up at the price of possible mistakes. Such heuristic strategies, called *strong rules* [19] reduce the computational cost using an active set strategy, but require difficult post-precessing to check for features possibly wrongly discarded. Another road to speed-up screening method has been the introduction of *sequential safe rules* [21, 23, 22]. The idea is to improve the screening thanks to the computations done for a previous regularization parameter. This scenario is particularly relevant in machine learning, where one computes solutions over a grid of regularization parameters, so as to select the best one (*e.g.,* to perform cross-validation). Nevertheless, such strategies suffer from the same problem as strong rules, since relevant features can be wrongly disregarded: sequential rules usually rely on theoretical quantities that are not known by the solver, but only approximated. Especially, for such rules to work one needs the exact dual optimal solution from the previous regularization parameter.

Recently, the introduction of *safe dynamic rules* [6, 5] has opened a promising venue by letting the screening to be done not only at the beginning of the algorithm, but all along the iterations. Following a method introduced for the Lasso [11], we generalize this dynamical safe rule, called GAP Safe rules (because it relies on duality gap computation) to a large class of learning problems with the following benefits:

- a unified and flexible framework for a wider family of problems,
- easy to insert in existing solvers,
- proved to be safe,
- more efficient that previous safe rules,
- achieves fast true active set identification.

We introduce our general GAP Safe framework in Section 2. We then specialize it to important machine learning use cases in Section 3. In Section 4 we apply our GAP Safe rules to a multi-task Lasso problem, relevant for brain imaging with magnetoencephalography data, as well as to multinomial logistic regression regularized with $\ell_1/\ell_2$ norm for joint feature selection.

## 2  GAP Safe rules

### 2.1  Model and notations

We denote by $[d]$ the set $\{1, \ldots, d\}$ for any integer $d \in \mathbb{N}$, and by $Q^\top$ the transpose of a matrix $Q$. Our observation matrix is $Y \in \mathbb{R}^{n \times q}$ where $n$ represents the number of samples, and $q$ the number of tasks or classes. The design matrix $X = [x^{(1)}, \ldots, x^{(p)}] = [x_1, \ldots, x_n]^\top \in \mathbb{R}^{n \times p}$ has $p$ explanatory variables (or features) column-wise, and $n$ observations row-wise. The standard $\ell_2$ norm is written $\|\cdot\|_2$, the $\ell_1$ norm $\|\cdot\|_1$, the $\ell_\infty$ norm $\|\cdot\|_\infty$. The $\ell_2$ unit ball is denoted by $\mathcal{B}_2$ (or simply $\mathcal{B}$) and we write $\mathcal{B}(c, r)$ the $\ell_2$ ball with center $c$ and radius $r$. For a matrix $B \in \mathbb{R}^{p \times q}$, we denote by $\|B\|_2^2 = \sum_{j=1}^p \sum_{k=1}^q B_{j,k}^2$ the Frobenius norm, and by $\langle \cdot, \cdot \rangle$ the associated inner product.

We consider the general optimization problem of minimizing a separable function with a group-Lasso regularization. The parameter to recover is a matrix $B \in \mathbb{R}^{p \times q}$, and for any $j$ in $\mathbb{R}^p$, $B_{j,:}$ is the $j$-th row of $B$, while for any $k$ in $\mathbb{R}^q$, $B_{:,k}$ is the $k$-th column. We would like to find

$$\widehat{B}^{(\lambda)} \in \underset{B \in \mathbb{R}^{p \times q}}{\arg \min} \underbrace{\sum_{i=1}^n f_i(x_i^\top B) + \lambda \Omega(B)}_{P_\lambda(B)} , \tag{1}$$

where $f_i : \mathbb{R}^{1 \times q} \mapsto \mathbb{R}$ is a convex function with $1/\gamma$-Lipschitz gradient. So $F : B \to \sum_{i=1}^n f_i(x_i^\top B)$ is also convex with Lipschitz gradient. The function $\Omega : \mathbb{R}^{p \times q} \mapsto \mathbb{R}_+$ is the $\ell_1/\ell_2$ norm $\Omega(B) = \sum_{j=1}^p \|B_{j,:}\|_2$ promoting a few lines of B to be non-zero at a time. The $\lambda$ parameter is a non-negative constant controlling the trade-off between data fitting and regularization.

Some elements of convex analysis used in the following are introduced here. For a convex function $f : \mathbb{R}^d \to [-\infty, +\infty]$ the Fenchel-Legendre transform[1] of $f$, is the function $f^* : \mathbb{R}^d \to [-\infty, +\infty]$ defined by $f^*(u) = \sup_{z \in \mathbb{R}^d} \langle z, u \rangle - f(z)$. The sub-differential of a function $f$ at a point $x$ is denoted by $\partial f(x)$. The dual norm of $\Omega$ is the $\ell_\infty/\ell_2$ norm and reads $\Omega_*(B) = \max_{j \in [p]} \|B_{j,:}\|_2$.

**Remark 1.** For the ease of reading, all groups are weighted with equal strength, but extension of our results to non-equal weights as proposed in the original group-Lasso [24] paper would be straightforward.

### 2.2  Basic properties

First we recall the associated Fermat's condition and a dual formulation of the optimization problem:

**Theorem 1.** ***Fermat's condition*** *(see [3, Proposition 26.1] for a more general result)*
*For any convex function* $f : \mathbb{R}^n \to \mathbb{R}$*:*

$$x \in \underset{x \in \mathbb{R}^n}{\arg \min} f(x) \Leftrightarrow 0 \in \partial f(x). \tag{2}$$

**Theorem 2** ([9]). *A dual formulation of* (1) *is given by*

$$\widehat{\Theta}^{(\lambda)} = \underset{\Theta \in \Delta_X}{\arg\max} - \underbrace{\sum_{i=1}^{n} f_i^*(-\lambda\Theta_{i,:})}_{D_\lambda(\Theta)} . \tag{3}$$

*where* $\Delta_X = \{\Theta \in \mathbb{R}^{n \times q} : \forall j \in [p], \|x^{(j)\top}\Theta\|_2 \leqslant 1\} = \{\Theta \in \mathbb{R}^{n \times q} : \Omega_*(X^\top\Theta) \leqslant 1\}$. *The primal and dual solutions are linked by*

$$\forall i \in [n], \quad \widehat{\Theta}_{i,:}^{(\lambda)} = -\nabla f_i(x_i^\top\widehat{B}^{(\lambda)})/\lambda. \tag{4}$$

*Furthermore, Fermat's condition reads:*

$$\forall j \in [p], \quad x^{(j)\top}\widehat{\Theta}^{(\lambda)} \in \begin{cases} \left\{ \frac{\widehat{B}_{j,:}^\lambda}{\|\widehat{B}_{j,:}^\lambda\|_2} \right\}, & \text{if } \widehat{B}_{j,:}^{(\lambda)} \neq 0, \\ \mathcal{B}_2, & \text{if } \widehat{B}_{j,:}^{(\lambda)} = 0. \end{cases} \tag{5}$$

**Remark 2.** Contrarily to the primal, the dual problem has a unique solution under our assumption on $f_i$. Indeed, the dual function is strongly concave, hence strictly concave.

**Remark 3.** For any $\Theta \in \mathbb{R}^{n \times q}$ let us introduce $G(\Theta) = [\nabla f_1(\Theta_{1,:})^\top, \ldots, \nabla f_n(\Theta_{n,:})^\top] \in \mathbb{R}^{n \times q}$. Then the primal/dual link can be written $\widehat{\Theta}^{(\lambda)} = -G(X\widehat{B}^{(\lambda)})/\lambda$ .

## 2.3 Critical parameter: $\lambda_{\max}$

For $\lambda$ large enough the solution of the primal problem is simply $0$. Thanks to the Fermat's rule (2), $0$ is optimal if and only if $-\nabla F(0)/\lambda \in \partial\Omega(0)$. Thanks to the property of the dual norm $\Omega_*$, this is equivalent to $\Omega_*(\nabla F(0)/\lambda) \leqslant 1$ where $\Omega_*$ is the dual norm of $\Omega$. Since $\nabla F(0) = X^\top G(0)$, $0$ is a primal solution of $P_\lambda$ if and only if $\lambda \geqslant \lambda_{\max} := \max_{j \in [p]} \|x^{(j)\top} G(0)\|_2 = \Omega_*(X^\top G(0))$.

This development shows that for $\lambda \geqslant \lambda_{\max}$, Problem (1) is trivial. So from now on, we will only focus on the case where $\lambda \leqslant \lambda_{\max}$.

## 2.4 Screening rules description

Safe screening rules rely on a simple consequence of the Fermat's condition:

$$\|x^{(j)\top}\widehat{\Theta}^{(\lambda)}\|_2 < 1 \Rightarrow \widehat{B}_{j,:}^{(\lambda)} = 0 . \tag{6}$$

Stated in such a way, this relation is useless because $\widehat{\Theta}^{(\lambda)}$ is unknown (unless $\lambda > \lambda_{\max}$). However, it is often possible to construct a set $\mathcal{R} \subset \mathbb{R}^{n \times q}$, called a *safe region*, containing it. Then, note that

$$\max_{\Theta \in \mathcal{R}} \|x^{(j)\top}\Theta\|_2 < 1 \Rightarrow \widehat{B}_{j,:}^{(\lambda)} = 0 . \tag{7}$$

The so called *safe screening rules* consist in removing the variable $j$ from the problem whenever the previous test is satisfied, since $\widehat{B}_{j,:}^{(\lambda)}$ is then guaranteed to be zero. This property leads to considerable speed-up in practice especially with active sets strategies, see for instance [11] for the Lasso case. A natural goal is to find safe regions as narrow as possible: smaller safe regions can only increase the number of screened out variables. However, complex regions could lead to a computational burden limiting the benefit of screening. Hence, we focus on constructing $\mathcal{R}$ satisfying the trade-off:

- $\mathcal{R}$ is as small as possible and contains $\widehat{\Theta}^{(\lambda)}$.
- Computing $\max_{\Theta \in \mathcal{R}} \|x^{(j)\top}\Theta\|_2$ is cheap.

## 2.5 Spheres as safe regions

Various shapes have been considered in practice for the set $\mathcal{R}$ such as balls (referred to as spheres) [9], domes [11] or more refined sets (see [23] for a survey). Here we consider the so-called "sphere regions" choosing a ball $\mathcal{R} = \mathcal{B}(c, r)$ as a safe region. One can easily obtain a control

on $\max_{\Theta \in \mathcal{B}(c,r)} \|x^{(j)\top}\Theta\|_2$ by extending the computation of the support function of a ball [11, Eq. (9)] to the matrix case: $\max_{\Theta \in \mathcal{B}(c,r)} \|x^{(j)\top}\Theta\|_2 \leqslant \|x^{(j)\top}c\|_2 + r\|x^{(j)}\|_2$ .

Note that here the center $c$ is a matrix in $\mathbb{R}^{p \times q}$. We can now state the safe sphere test:

$$\text{Sphere test:} \qquad \text{If} \quad \|x^{(j)\top}c\|_2 + r\|x^{(j)}\|_2 < 1, \quad \text{then} \quad \widehat{B}_{j,:}^{(\lambda)} = 0. \tag{8}$$

### 2.6 GAP Safe rule description

In this section we derive a GAP Safe screening rule extending the one introduced in [11]. For this, we rely on the strong convexity of the dual objective function and on weak duality.

**Finding a radius:** Remember that $\forall i \in [n], f_i$ is differentiable with a $1/\gamma$-Lipschitz gradient. As a consequence, $\forall i \in [n], f_i^*$ is $\gamma$-strongly convex [14, Theorem 4.2.2, p. 83] and so $D_\lambda$ is $\gamma\lambda^2$-strongly concave:

$$\forall (\Theta_1, \Theta_2) \in \mathbb{R}^{n \times q} \times \mathbb{R}^{n \times q}, \quad D_\lambda(\Theta_2) \leqslant D_\lambda(\Theta_1) + \langle \nabla D_\lambda(\Theta_1), \Theta_2 - \Theta_1 \rangle - \frac{\gamma\lambda^2}{2}\|\Theta_1 - \Theta_2\|^2.$$

Specifying the previous inequality for $\Theta_1 = \widehat{\Theta}^{(\lambda)}, \Theta_2 = \Theta \in \Delta_X$, one has

$$D_\lambda(\Theta) \leqslant D_\lambda(\widehat{\Theta}^{(\lambda)}) + \langle \nabla D_\lambda(\widehat{\Theta}^{(\lambda)}), \Theta - \widehat{\Theta}^{(\lambda)} \rangle - \frac{\gamma\lambda^2}{2}\|\widehat{\Theta}^{(\lambda)} - \Theta\|^2.$$

By definition, $\widehat{\Theta}^{(\lambda)}$ maximizes $D_\lambda$ on $\Delta_X$, so we have: $\langle \nabla D_\lambda(\widehat{\Theta}^{(\lambda)}), \Theta - \widehat{\Theta}^{(\lambda)} \rangle \leqslant 0$. This implies

$$D_\lambda(\Theta) \leqslant D_\lambda(\widehat{\Theta}^{(\lambda)}) - \frac{\gamma\lambda^2}{2}\|\widehat{\Theta}^{(\lambda)} - \Theta\|^2.$$

By weak duality $\forall B \in \mathbb{R}^{p \times q}, D_\lambda(\widehat{\Theta}^{(\lambda)}) \leqslant P_\lambda(B)$, so : $\forall B \in \mathbb{R}^{p \times q}, \forall \Theta \in \Delta_X, D_\lambda(\Theta) \leqslant P_\lambda(B) - \frac{\gamma\lambda^2}{2}\|\widehat{\Theta}^{(\lambda)} - \Theta\|^2$, and we deduce the following theorem:

**Theorem 3.**
$$\forall B \in \mathbb{R}^{p \times q}, \forall \Theta \in \Delta_X, \quad \left\|\widehat{\Theta}^{(\lambda)} - \Theta\right\|_2 \leqslant \sqrt{\frac{2(P_\lambda(B) - D_\lambda(\Theta))}{\gamma\lambda^2}} =: \hat{r}_\lambda(B, \Theta). \tag{9}$$

Provided one knows a dual feasible point $\Theta \in \Delta_X$ and a $B \in \mathbb{R}^{p \times q}$, it is possible to construct a safe sphere with radius $\hat{r}_\lambda(B, \Theta)$ centered on $\Theta$. We now only need to build a (relevant) dual point to center such a ball. Results from Section 2.3, ensure that $-G(0)/\lambda_{\max} \in \Delta_X$, but it leads to a static rule, a introduced in [9]. We need a dynamic center to improve the screening as the solver proceeds.

**Finding a center:** Remember that $\widehat{\Theta}^{(\lambda)} = -G(X\widehat{B}^{(\lambda)})/\lambda$. Now assume that one has a converging algorithm for the primal problem, *i.e.,* $B_k \to \widehat{B}^{(\lambda)}$. Hence, a natural choice for creating a dual feasible point $\Theta_k$ is to choose it proportional to $-G(XB_k)$, for instance by setting:

$$\Theta_k = \begin{cases} \frac{R_k}{\lambda}, & \text{if } \Omega_*(X^\top R_k) \leqslant \lambda, \\ \frac{R_k}{\Omega_*(X^\top R_k)}, & \text{otherwise.} \end{cases} \qquad \text{where } R_k = -G(XB_k) . \tag{10}$$

A refined method consists in solving the one dimensional problem: $\arg\max_{\Theta \in \Delta_X \cap \text{Span}(R_k)} D_\lambda(\Theta)$. In the Lasso and group-Lasso case [5, 6, 11] such a step is simply a projection on the intersection of a line and the (polytope) dual set and can be computed efficiently. However for logistic regression the computation is more involved, so we have opted for the simpler solution in Equation (10). This still provides converging safe rules (see Proposition 1).

**Dynamic GAP Safe rule summarized**

We can now state our dynamical GAP Safe rule at the $k$-th step of an iterative solver:

    1. Compute $B_k$, and then obtain $\Theta_k$ and $\hat{r}_\lambda(B_k, \Theta_k)$ using (10).

2. If $\|x^{(j)^\top}\Theta_k\|_2 + \hat{r}_\lambda(\mathrm{B}_k,\Theta_k)\|x^{(j)}\|_2 < 1,$ then set $\widehat{\mathrm{B}}^{(\lambda)}_{j,:} = 0$ and remove $x^{(j)}$ from $X$.

Dynamic safe screening rules are more efficient than existing methods in practice because they can increase the ability of screening as the algorithm proceeds. Since one has sharper and sharper dual regions available along the iterations, support identification is improved. Provided one relies on a primal converging algorithm, one can show that the dual sequence we propose is converging too.

The convergence of the primal is unaltered by our GAP Safe rule: screening out unnecessary coefficients of $\mathrm{B}_k$ can only decrease its distance with its original limits. Moreover, a practical consequence is that one can observe surprising situations where lowering the tolerance of the solver can reduce the computation time. This can happen for sequential setups.

**Proposition 1.** *Let* $\mathrm{B}_k$ *be the current estimate of* $\widehat{\mathrm{B}}^{(\lambda)}$ *and* $\Theta_k$ *defined in Eq.* (10) *be the current estimate of* $\widehat{\Theta}^{(\lambda)}$. *Then* $\lim_{k\to+\infty}\mathrm{B}_k = \widehat{\mathrm{B}}^{(\lambda)}$ *implies* $\lim_{k\to+\infty}\Theta_k = \widehat{\Theta}^{(\lambda)}$.

Note that if the primal sequence is converging to the optimal, our dual sequence is also converging. But we know that the radius of our safe sphere is $(2(P_\lambda(\mathrm{B}_k) - D_\lambda(\Theta_k))/(\gamma\lambda^2))^{1/2}$. By strong duality, this radius converges to 0, hence we have certified that our GAP Safe regions sequence $\mathcal{B}(\Theta_k, \hat{r}_\lambda(\mathrm{B}_k,\Theta_k))$ is a converging safe rules (in the sense introduced in [11, Definition 1]).

**Remark 4.** The active set obtained by our GAP Safe rule (*i.e.,* the indexes of non screened-out variables) converges to the equicorrelation set [20] $\mathcal{E}_\lambda := \{j \in p : \|x^{(j)^\top}\widehat{\Theta}^{(\lambda)}\|_2 = 1\}$, allowing us to early identify relevant features (see Proposition 2 in the supplementary material for more details).

## 3 Special cases of interest

We now specialize our results to relevant supervised learning problems, see also Table 1.

### 3.1 Lasso

In the Lasso case $q = 1$, the parameter is a vector: $\mathrm{B} = \beta \in \mathbb{R}^p$, $F(\beta) = 1/2\|y - X\beta\|_2^2 = \sum_{i=1}^n (y_i - x_i^\top\beta)^2$, meaning that $f_i(z) = (y_i - z)^2/2$ and $\Omega(\beta) = \|\beta\|_1$.

### 3.2 $\ell_1/\ell_2$ multi-task regression

In the multi-task Lasso, which is a special case of group-Lasso, we assume that the observation is $Y \in \mathbb{R}^{n\times q}$, $F(\mathrm{B}) = \frac{1}{2}\|Y - X\mathrm{B}\|_2^2 = \frac{1}{2}\sum_{i=1}^n \|Y_{i,:} - x_i^\top\mathrm{B}\|_2^2$ (*i.e.,* $f_i(z) = \|Y_{i,:} - z\|^2/2$) and $\Omega(\mathrm{B}) = \sum_{j=1}^p \|\mathrm{B}_{j,:}\|_2$. In signal processing, this model is also referred to as Multiple Measurement Vector (MMV) problem. It allows to jointly select the same features for multiple regression tasks [1, 2].

**Remark 5.** Our framework could encompass easily the case of non-overlapping groups with various size and weights presented in [6]. Since our aim is mostly for multi-task and multinomial applications, we have rather presented a matrix formulation.

### 3.3 $\ell_1$ regularized logistic regression

Here, we consider the formulation given in [7, Chapter 3] for the two classes logistic regression. In such a context, one observes for each $i \in [n]$ a class label $c_i \in \{1, 2\}$. This information can be recast as $y_i = \mathbb{1}_{\{c_i=1\}}$, and it is then customary to minimize (1) where

$$F(\beta) = \sum_{i=1}^n \left(-y_i x_i^\top\beta + \log\left(1 + \exp\left(x_i^\top\beta\right)\right)\right), \tag{11}$$

with $\mathrm{B} = \beta \in \mathbb{R}^p$ (*i.e., $q = 1$*), $f_i(z) = -y_i z + \log(1 + \exp(z))$ and the penalty is simply the $\ell_1$ norm: $\Omega(\beta) = \|\beta\|_1$. Let us introduce Nh, the (binary) negative entropy function defined by [2]:

$$\mathrm{Nh}(x) = \begin{cases} x\log(x) + (1-x)\log(1-x), & \text{if } x \in [0,1] \ , \\ +\infty, & \text{otherwise} \ . \end{cases} \tag{12}$$

Then, one can easily check that $f_i^*(z_i) = \mathrm{Nh}(z_i + y_i)$ and $\gamma = 4$.

|  | Lasso | Multi-task regr. | Logistic regr. | Multinomial regr. |
|---|---|---|---|---|
| $f_i(z)$ | $\frac{(y_i-z)^2}{2}$ | $\frac{\|Y_{i,:}-z\|^2}{2}$ | $\log(1+\mathrm{e}^z)-y_i z$ | $\log\left(\sum_{k=1}^{q}\mathrm{e}^{z_k}\right)-\sum_{k=1}^{q}Y_{i,k}z_k$ |
| $f_i^*(u)$ | $\frac{(y_i-u)^2-y_i^2}{2}$ | $\frac{\|Y_{i,:}-u\|^2-\|Y_{i,:}\|_2^2}{2}$ | $\mathrm{Nh}(u+y_i)$ | $\mathrm{NH}(u+Y_{i,:})$ |
| $\Omega(\mathrm{B})$ | $\|\beta\|_1$ | $\sum_{j=1}^{p}\|\mathrm{B}_{j,:}\|_2$ | $\|\beta\|_1$ | $\sum_{j=1}^{p}\|\mathrm{B}_{j,:}\|_2$ |
| $\lambda_{\max}$ | $\|X^\top y\|_\infty$ | $\Omega_*(X^\top Y)$ | $\|X^\top(\mathbf{1}_n/2-y)\|_\infty$ | $\Omega_*(X^\top(\mathbf{1}_{n\times q}/q-Y))$ |
| $G(\Theta)$ | $\theta-y$ | $\Theta-Y$ | $\frac{\mathrm{e}^z}{1+\mathrm{e}^z}-y$ | $\mathrm{RowNorm}(\mathrm{e}^\Theta)-Y$ |
| $\gamma$ | 1 | 1 | 4 | 1 |

Table 1: Useful ingredients for computing GAP Safe rules. We have used lower case to indicate when the parameters are vectorial (*i.e., q = 1*). The function RowNorm consists in normalizing a (non-negative) matrix row-wise, such that each row sums to one.

### 3.4 $\ell_1/\ell_2$ multinomial logistic regression

We adapt the formulation given in [7, Chapter 3] for the multinomial regression. In such a context, one observes for each $i \in [n]$ a class label $c_i \in \{1, \ldots, q\}$. This information can be recast into a matrix $Y \in \mathbb{R}^{n\times q}$ filled by 0's and 1's: $Y_{i,k} = \mathbb{1}_{\{c_i=k\}}$. In the same spirit as the multi-task Lasso, a matrix $\mathrm{B} \in \mathbb{R}^{p\times q}$ is formed by $q$ vectors encoding the hyperplanes for the linear classification. The multinomial $\ell_1/\ell_2$ regularized regression reads:

$$F(\mathrm{B}) = \sum_{i=1}^{n}\left(\sum_{k=1}^{q}-Y_{i,k}x_i^\top \mathrm{B}_{:,k}+\log\left(\sum_{k=1}^{q}\exp\left(x_i^\top \mathrm{B}_{:,k}\right)\right)\right), \qquad (13)$$

with $f_i(z) = \sum_{k=1}^{q}-Y_{i,k}z_k + \log\left(\sum_{k=1}^{q}\exp\left(z_k\right)\right)$ to recover the formulation as in (1). Let us introduce NH, the negative entropy function defined by (still with the convention $0\log(0)=0$)

$$\mathrm{NH}(x) = \begin{cases} \sum_{i=1}^{q}x_i\log(x_i), & \text{if } x\in\Sigma_q=\{x\in\mathbb{R}_+^q:\sum_{i=1}^{q}x_i=1\}, \\ +\infty, & \text{otherwise.} \end{cases} \qquad (14)$$

Again, one can easily check that $f_i^*(z) = \mathrm{NH}(z+Y_{i,:})$ and $\gamma = 1$.

**Remark 6.** For multinomial logistic regression, $D_\lambda$ implicitly encodes the additional constraint $\Theta \in \mathrm{dom}\, D_\lambda = \{\Theta' : \forall i\in[n], -\lambda\Theta'_{i,:}+Y_{i,:}\in\Sigma_q\}$ where $\Sigma_q$ is the $q$ dimensional simplex, see (14). As 0 and $R_k/\lambda$ both belong to this set, any convex combination of them, such as $\Theta_k$ defined in (10), satisfies this additional constraint.

**Remark 7.** The intercept has been neglected in our models for simplicity. Our GAP Safe framework can also handle such a feature at the cost of more technical details (by adapting the results from [15] for instance). However, in practice, the intercept can be handled in the present formulation by adding a constant column to the design matrix $X$. The intercept is then regularized. However, if the constant is set high enough, regularization is small and experiments show that it has little to no impact for high-dimensional problems. This is the strategy used by the Liblinear package [10].

## 4 Experiments

In this section we present results obtained with the GAP Safe rule. Results are on high dimensional data, both dense and sparse. Implementation have been done in Python and Cython for low critical parts. They are based on the multi-task Lasso implementation of Scikit-Learn [17] and coordinate descent logistic regression solver in the Lightning software [4]. In all experiments, the coordinate descent algorithm used follows the pseudo code from [11] with a screening step every 10 iterations.

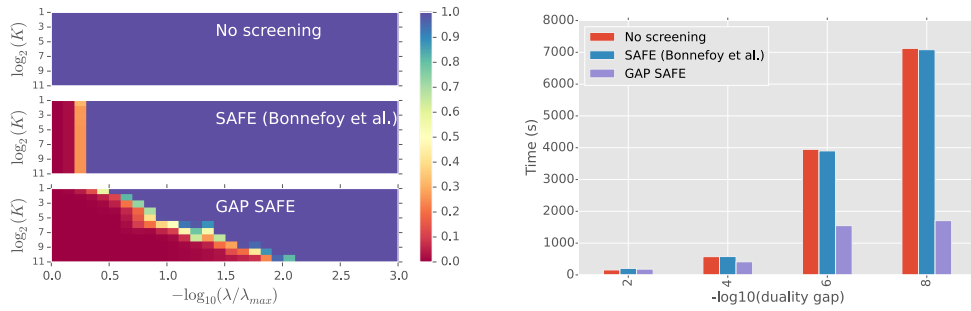

Figure 1: Experiments on MEG/EEG brain imaging dataset (dense data with $n = 360$, $p = 22494$ and $q = 20$). On the left: fraction of active variables as a function of $\lambda$ and the number of iterations $K$. The GAP Safe strategy has a much longer range of $\lambda$ with (red) small active sets. On the right: Computation time to reach convergence using different screening strategies.

Note that we have not performed comparison with the sequential screening rule commonly acknowledge as the state-of-the-art "safe" screening rule (such as th EDDP+ [21]), since we can show that this kind of rule is not safe. Indeed, the stopping criterion is based on dual gap accuracy, and comparisons would be unfair since such methods sometimes do not converge to the prescribed accuracy. This is backed-up by a counter example given in the supplementary material. Nevertheless, modifications of such rules, inspired by our GAP Safe rules, can make them safe. However the obtained sequential rules are still outperformed by our dynamic strategies (see Figure 2 for an illustration).

## 4.1  $\ell_1/\ell_2$ multi-task regression

To demonstrate the benefit of the GAP Safe screening rule for a multi-task Lasso problem we used neuroimaging data. Electroencephalography (EEG) and magnetoencephalography (MEG) are brain imaging modalities that allow to identify active brain regions. The problem to solve is a multi-task regression problem with squared loss where every task corresponds to a time instant. Using a multi-task Lasso one can constrain the recovered sources to be identical during a short time interval [13]. This corresponds to a temporal stationary assumption. In this experiment we used a joint MEG/EEG data with 301 MEG and 59 EEG sensors leading to $n = 360$. The number of possible sources is $p = 22,494$ and the number of time instants $q = 20$. With a 1 kHz sampling rate it is equivalent to say that the sources stay the same for 20 ms.

Results are presented in Figure 1. The GAP Safe rule is compared with the dynamic safe rule from [6]. The experimental setup consists in estimating the solutions of the multi-task Lasso problem for 100 values of $\lambda$ on a logarithmic grid from $\lambda_{\max}$ to $\lambda_{\max}/10^3$. For the experiments on the left a fixed number of iterations from 2 to $2^{11}$ is allowed for each $\lambda$. The fraction of active variables is reported. Figure 1 illustrates that the GAP Safe rule screens out much more variables than the compared method, as well as the converging nature of our safe regions. Indeed, the more iterations performed the more the rule allows to screen variables. On the right, computation time confirms the effective speed-up. Our rule significantly improves the computation time for all duality gap tolerance from $10^{-2}$ to $10^{-8}$, especially when accurate estimates are required, *e.g.,* for feature selection.

## 4.2  $\ell_1$ binary logistic regression

Results on the Leukemia dataset are reported in Figure 2. We compare the dynamic strategy of GAP Safe to a sequential and non dynamic rule such as Slores [22]. We do not compare to the actual Slores rule as it requires the previous dual optimal solution, which is not available. Slores is indeed not a safe method (see Section B in the supplementary materials). Nevertheless one can observe that dynamic strategies outperform pure sequential one, see Section C in the supplementary material).

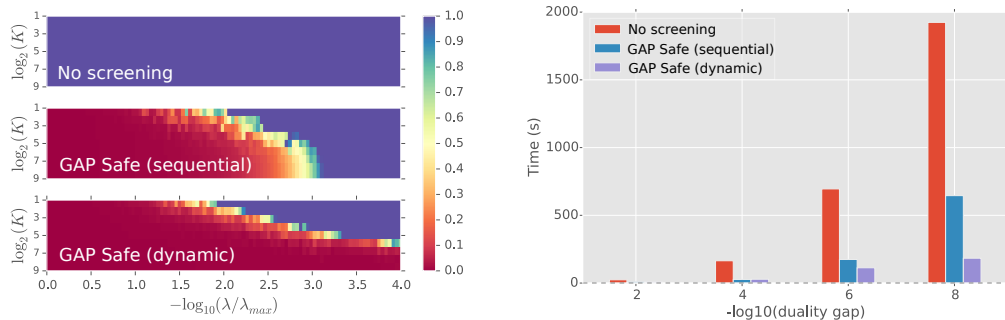

Figure 2: $\ell_1$ regularized binary logistic regression on the Leukemia dataset ($n = 72$ ; $m = 7,129$ ; $q = 1$). Simple sequential and full dynamic screening GAP Safe rules are compared. On the left: fraction of the variables that are active. Each line corresponds to a fixed number of iterations for which the algorithm is run. On the right: computation times needed to solve the logistic regression path to desired accuracy with 100 values of $\lambda$.

### 4.3 $\ell_1/\ell_2$ multinomial logistic regression

We also applied GAP Safe to an $\ell_1/\ell_2$ multinomial logistic regression problem on a sparse dataset. Data are bag of words features extracted from the News20 dataset (TF-IDF removing English stop words and words occurring only once or more than 95% of the time). One can observe on Figure 3 the dynamic screening and its benefit as more iterations are performed. GAP Safe leads to a significant speedup: to get a duality gap smaller than $10^{-2}$ on the 100 values of $\lambda$, we needed 1,353 s without screening and only 485 s when GAP Safe was activated.

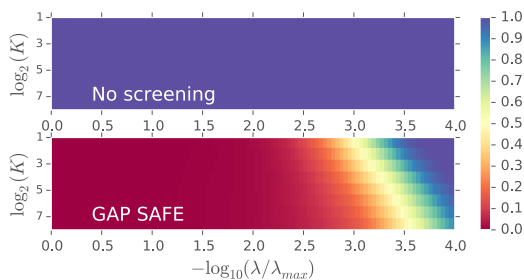

Figure 3: Fraction of the variables that are active for $\ell_1/\ell_2$ regularized multinomial logistic regression on 3 classes of the News20 dataset (sparse data with $n = 2,757$ ; $m = 13,010$ ; $q = 3$). Computation was run on the best 10% of the features using $\chi^2$ univariate feature selection [16]. Each line corresponds to a fixed number of iterations for which the algorithm is run.

## 5 Conclusion

This contribution detailed new safe rules for accelerating algorithms solving generalized linear models regularized with $\ell_1$ and $\ell_1/\ell_2$ norms. The rules proposed are safe, easy to implement, dynamic and converging, allowing to discard significantly more variables than alternative safe rules. The positive impact in terms of computation time was observed on all tested datasets and demonstrated here on a high dimensional regression task using brain imaging data as well as binary and multiclass classification problems on dense and sparse data. Extensions to other generalized linear model, *e.g.,* Poisson regression, are expected to reach the same conclusion. Future work could investigate optimal screening frequency, determining when the screening has correctly detected the support.

## Acknowledgment

We acknowledge the support from Chair Machine Learning for Big Data at Télécom ParisTech and from the Orange/Télécom ParisTech think tank phi-TAB. This work benefited from the support of the "FMJH Program Gaspard Monge in optimization and operation research", and from the support to this program from EDF.

## Footnotes

[1]this is also often referred to as the (convex) conjugate of a function

[2]with the convention $0\log(0) = 0$

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
