[Supplementary Material · nips2015_supp.pdf]

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

## Supplementary Material

## A  Proofs

### A.1  Proof of variable identification

**Proposition 2.** *There exists $k_0 \in \mathbb{N}$ such that for all $k \geqslant k_0$, an index $j \in [p]$ is screened out by the GAP Safe rule if and only if $j \in \mathcal{E}_\lambda := \{j \in p : \|x^{(j)}{}^\top \widehat{\Theta}^{(\lambda)}\|_2 = 1\}$.*

*Proof.* For simplicity we use the notation $\mathcal{R}_k = \mathcal{B}(\Theta_k, \hat{r}_\lambda(\mathrm{B}_k, \Theta_k))$ for the safe region at step $k$. Define $\max_{j \notin \mathcal{E}_\lambda} |x^{(j)}{}^\top \widehat{\Theta}^{(\lambda)}| = t < 1$. Fix $\epsilon > 0$ such that $\epsilon < (1 - t)/(\max_{j \notin \mathcal{E}_\lambda} \|x^{(j)}\|)$. As $\Theta_k$ is converging to $\widehat{\Theta}^{(\lambda)}$, and $\lim_{k \to \infty} \hat{r}_\lambda(\mathrm{B}_k, \Theta_k) = 0$, there exists $k_0 \in \mathbb{N}$ such that $\forall k \geqslant k_0, \forall \Theta \in \mathcal{R}_k, \|\Theta - \widehat{\Theta}^{(\lambda)}\| \leqslant \epsilon$. Hence, for any $j \notin \mathcal{E}_\lambda$ and any $\Theta \in \mathcal{R}_k$, $|x^{(j)}{}^\top (\Theta - \widehat{\Theta}^{(\lambda)})| \leqslant (\max_{j \notin \mathcal{E}_\lambda} \|x^{(j)}\|)\|\Theta - \widehat{\Theta}^{(\lambda)}\| \leqslant (\max_{j \notin \mathcal{E}_\lambda} \|x^{(j)}\|)\epsilon$. Using the triangle inequality, one gets

$$|x^{(j)}{}^\top \Theta| \leqslant (\max_{j \notin \mathcal{E}_\lambda} \|x^{(j)}\|)\epsilon + \max_{j \notin \mathcal{E}_\lambda} |x^{(j)}{}^\top \widehat{\Theta}^{(\lambda)}|$$
$$\leqslant (\max_{j \notin \mathcal{E}_\lambda} \|x^{(j)}\|)\epsilon + t < 1,$$

provided that $\epsilon < (1 - t)/(\max_{j \notin \mathcal{E}_\lambda} \|x^{(j)}\|)$. Hence, for all $k \geqslant k_0, j \notin \mathcal{E}_\lambda$ implies that $j$ is screened out by the GAP Safe rule thanks to the last inequality. For the reverse inclusion take $j \in \mathcal{E}_\lambda$, *i.e.*, $|x^{(j)}{}^\top \widehat{\Theta}^{(\lambda)}| = 1$. Since by construction of our GAP Safe screening rule $\forall k \in \mathbb{N}, \widehat{\Theta}^{(\lambda)} \in \mathcal{R}_k$, then $j \in \{j' \in [p] : \max_{\Theta \in \mathcal{R}_k} |x^{(j')}{}^\top \Theta| \geqslant 1\}$. This means that the variable $j$ can not be eliminated by our safe rule, and we have shown that in the limit we have exactly identified the equicorrelation set. $\qquad\square$

### A.2  Proof that the GAP Safe rule is converging (Proposition 1)

*Proof.* We consider two cases.

First let us assume that $\theta_k = R_k/\Omega_*(X^\top G(X\mathrm{B}_k))$

$$\left\|\Theta_k - \widehat{\Theta}^{(\lambda)}\right\|_2 = \left\|\frac{-G(X\mathrm{B}_k)}{\Omega_*(X^\top G(X\mathrm{B}_k))} + \frac{1}{\lambda}G(X\widehat{\mathrm{B}}^{(\lambda)})\right\|_2$$
$$\leqslant \left\|\frac{G(X\mathrm{B}_k)}{\lambda} - \frac{G(X\mathrm{B}_k)}{\Omega_*(X^\top G(X\mathrm{B}_k))}\right\|_2 + \left\|\frac{G(X\widehat{\mathrm{B}}^{(\lambda)}) - G(X\mathrm{B}_k)}{\lambda}\right\|_2$$
$$\leqslant \left|\frac{1}{\lambda} - \frac{1}{\Omega_*(X^\top G(X\mathrm{B}_k))}\right| \|G(X\mathrm{B}_k)\|_2 + \left\|\frac{G(X\widehat{\mathrm{B}}^{(\lambda)}) - G(X\mathrm{B}_k)}{\lambda}\right\|_2$$

The second term converges to zero whenever $\mathrm{B}_k \to \widehat{\mathrm{B}}^{(\lambda)}$ since $G$ is continuous (it is $\gamma$-Lipschitz). For the first term, note that $\Omega_*(X^\top G(X\mathrm{B}_k)) \to \Omega_*(X^\top G(X\widehat{\mathrm{B}}^{(\lambda)})) = \lambda\Omega_*(X^\top \widehat{\Theta}^{(\lambda)}) = \lambda$ (thanks to the primal/dual link, and that $\widehat{\Theta}^{(\lambda)}$ is dual feasible). Then, as $G$ is a Lipschitz function and all norms are equivalent in a finite dimension space, the right hand side converges to zero in the previous inequality, and the results stated follows.

In the second case $\Theta_k = R_k/\lambda$, so $\left\|\Theta_k - \widehat{\Theta}^{(\lambda)}\right\|_2 = \left\|\frac{-G(X\mathrm{B}_k)+G(X\widehat{\mathrm{B}}^{(\lambda)})}{\lambda}\right\|_2$ and the proof proceeds as in the first case.

$\qquad\square$

## B  EDPP is not safe

In the two last sections, we present a study on the EDDP method [21], a screening rule that relies on the dual optimal point obtained for the previous $\lambda$ in the path. Note that the same conclusion

would hold true for generalization of the sequential approach given in [22], as well as for any other screening rule that needs exact dual solution at one step. To simplify the reading we use the vectorial (with no capital letters) notation used earlier. In the remainder we consider $\lambda_0 = \lambda_{\max}$ and a non-increasing sequence of $T - 1$ tuning parameters $(\lambda_t)_{t \in [T-1]}$ in $(0, \lambda_{\max})$. In practice, we choose the common grid [7][2.12.1]: $\lambda_t = \lambda_0 10^{-\delta t/(T-1)}$. Wang *et al.* [21] proposed a sequential screening rule based on properties of the projection onto a convex set. Their rule is based on the exact knowledge of the true optimal solution for the previous parameter. Such a rule can be used to compute $\hat{\theta}^{(\lambda_1)}$ since $\hat{\theta}^{(\lambda_0)} = y/\lambda_0 \,(= y/\lambda_{\max})$ is known. However for $t > 1$, $\hat{\theta}^{(\lambda_t)}$ is only known approximately and the rules introduced in [21] are not safe anymore: some active groups may be wrongly disregarded if one does not use the exact value of $\hat{\theta}^{(\lambda_t)}$.

We first first recall the property they proved. Then, we give a counter-example that shows that the rule is indeed not safe. In Section C, we propose to modify their rule in order to make it safe in all cases.

Recall that in this case $q = 1$, the parameters are vectors: $\mathrm{B} = \beta \in \mathbb{R}^p$ and $\Theta = \theta \in \mathbb{R}^n$.

**Proposition 3** ([21, Theorem 19]). *Assume that $\lambda_{t-1} < \lambda_{\max}$, then the dual optimal solution of the group-Lasso with parameter $\lambda_t$, satisfies*

$$\hat{\theta}^{(\lambda_t)} \in \mathcal{B}\big(\hat{\theta}^{(\lambda_{t-1})} + \frac{1}{2}v^{\perp}(\lambda_{t-1}, \lambda_t), \frac{1}{2}\big\|v^{\perp}(\lambda_{t-1}, \lambda_t)\big\|_2\big) \qquad (15)$$

*where*

$$v^{\perp}(\lambda_{t-1}, \lambda_t) = \frac{y}{\lambda_t} - \hat{\theta}^{(\lambda_{t-1})} - \alpha[\hat{\theta}^{(\lambda_{t-1})}]\big(\frac{y}{\lambda_{t-1}} - \hat{\theta}^{(\lambda_{t-1})}\big)$$

*and*

$$\alpha[\hat{\theta}^{(\lambda_{t-1})}] := \underset{\alpha \in \mathbb{R}_+}{\arg\min} \left\| \frac{y}{\lambda_t} - \hat{\theta}^{(\lambda_{t-1})} - \alpha\big(\frac{y}{\lambda_{t-1}} - \hat{\theta}^{(\lambda_{t-1})}\big)\right\|_2$$

$$= \frac{\big\langle \frac{y}{\lambda_{t-1}} - \hat{\theta}^{(\lambda_{t-1})}, \frac{y}{\lambda_t} - \hat{\theta}^{(\lambda_{t-1})}\big\rangle}{\big\|\frac{y}{\lambda_{t-1}} - \hat{\theta}^{(\lambda_{t-1})}\big\|_2^2}. \qquad (16)$$

Note that the rule proposed by [21] (as pointed out in [6]) relies on the exact knowledge of a dual optimal solution for a previously solved Lasso problem. This is impossible to obtain in practice and even if it is possible to find accurate solutions, the search for high accuracy may hinder the benefits of the screening when it was not actually needed. Using inaccurate solutions may lead to discarding variables that should have been active and so the screened optimization algorithm will not converge to a solution of the original problem.

We illustrate this issue on Figure 4. Knowing an approximation $\beta$ to the optimal primal point, returned by the optimization algorithm at the previous regularization parameter $\lambda_{t-1}$, we need to choose an approximation $\theta$ to the optimal dual point to run EDPP.

- If we choose to approximate the dual optimal point by $\theta = \frac{1}{\lambda_{t-1}}(y - X\beta)$ (blue curve with diamonds), then the result is catastrophic. Indeed, at $\lambda_1$, $\beta = 0$ is a valid $\epsilon$-solution for $\epsilon = 10^{-1.5}$ and the screening rule tries to perform a division by 0 when computing $\alpha[\theta]$.
- If we choose to approximate the dual optimal point by $\frac{1}{\max(\lambda_{t-1}, \|X^{\top}(y-X\beta)\|_{\infty})}(y - X\beta)$, we have a better behavior (purple curve with triangles) but we may still have an algorithm which does not converge to an $\epsilon$-solution. Here, for the 13th Lasso problem a variable is erroneously removed and the problem can only be solved to accuracy $0.03515 > 10^{-1.5} \approx 0.03162$. This may look like a small issue but when the stopping criterion is based on the duality gap, this causes the algorithm to continue until the maximum number of iterations is reached.

## C  Making EDDP screening rule safe

### C.1  The simpler screening rule

In the present paper, we give computable guarantees on the distance between the current dual feasible point and the solution of the problem. We show here how we can combine our result with Wang

$$X = \begin{bmatrix} 1/\sqrt{2} & \sqrt{2}/\sqrt{3} \\ 0 & -1/\sqrt{6} \\ -1/\sqrt{2} & -1/\sqrt{6} \end{bmatrix}, \qquad y = \begin{bmatrix} 1/\sqrt{6} \\ 1/\sqrt{6} \\ -\sqrt{2}/\sqrt{3} \end{bmatrix}.$$

Figure 4: EDPP is not safe. We run GAP SAFE and two interpretations of EDPP (described in the main text) to solve the Lasso path on the dataset defined by $X$ and $y$ above with target accuracy $10^{-1.5}$. For each Lasso problem, we plot the final duality gap returned by the optimization solver.

*et al.* 's in order to make their screening rule work even with approximate solutions to the previous Lasso problem.

For simplicity, we first consider the initial version of Wang *et al.* 's sphere test:

$$\hat{\theta}^{(\lambda_t)} \in \mathcal{B}\big(\hat{\theta}^{(\lambda_{t-1})}, \big\| v^{\perp}(\lambda_{t-1}, \lambda_t) \big\|_2\big), \tag{17}$$

proved in [21, Theorem 7]. As we do not know $\hat{\theta}^{(\lambda_{t-1})}$, we cannot readily use this ball. However, we can modify it to make it a safe screening rules as follows:

**Proposition 4.** *Assume that* $\lambda_{t-1} < \lambda_{\max}$, $\theta \in \Delta_X$ *is a dual feasible point and* $r_{\lambda_{t-1}} > 0$ *is a radius satisfying* $\hat{\theta}^{(\lambda_{t-1})} \in \mathcal{B}(\theta, r_{\lambda_{t-1}})$, *then*

$$\hat{\theta}^{(\lambda_t)} \in \mathcal{B}\Big(\theta, r_{\lambda_{t-1}}(1 + |1 - \alpha[\theta]|) + \Big\| \frac{y}{\lambda_t} - \theta - \alpha[\theta](\frac{y}{\lambda_{t-1}} - \theta) \Big\|_2\Big), \tag{18}$$

*where*

$$\alpha[\theta] := \underset{\alpha \in \mathbb{R}_+}{\arg\min} \Big\| \frac{y}{\lambda_t} - \theta - \alpha(\frac{y}{\lambda_{t-1}} - \theta) \Big\|_2 = \left( \frac{\langle \frac{y}{\lambda_{t-1}} - \theta, \frac{y}{\lambda_t} - \theta \rangle}{\| \frac{y}{\lambda_{t-1}} - \theta \|_2^2} \right)_+, \tag{19}$$

*and for any* $t \in \mathbb{R}$, $(t)_+ = \max(0, t)$.

*Proof.* Start first by noting that (17) implies

$$\hat{\theta}^{(\lambda_t)} \in \bigcup_{\theta' \in \mathcal{B}(\theta, r_{\lambda_{t-1}})} \mathcal{B}\Big(\theta', \min_{\alpha \in \mathbb{R}_+} \Big\| \frac{y}{\lambda_t} - \theta' - \alpha(\frac{y}{\lambda_{t-1}} - \theta') \Big\|_2\Big).$$

Let us denote

$$H = \max_{\theta' \in \mathcal{B}(\theta, r_{\lambda_{t-1}})} \min_{\alpha \in \mathbb{R}_+} \Big\| \frac{y}{\lambda_t} - \theta' - \alpha(\frac{y}{\lambda_{t-1}} - \theta') \Big\|_2,$$

then $\hat{\theta}^{(\lambda_t)} \in \mathcal{B}(\theta, r_{\lambda_{t-1}} + H)$. We now need to upper bound $H$. A simple choice is to take $\alpha$ to be $\alpha[\theta]$ defined in Eq. (19) The motivation for such a choice is because it is optimal when $r_{\lambda_{t-1}} = 0$. This provides the following bound on $H$:

$$H \leqslant \max_{\theta' \in \mathcal{B}(\theta, r_{\lambda_{t-1}})} \Big\| \frac{y}{\lambda_t} - \theta' - \alpha[\theta](\frac{y}{\lambda_{t-1}} - \theta') \Big\|_2,$$

$$= \left\| \frac{y}{\lambda_t} - \theta - \alpha[\theta](\frac{y}{\lambda_{t-1}} - \theta) + r_{\lambda_{t-1}}(\alpha[\theta] - 1) \frac{\frac{y}{\lambda_t} - \theta - \alpha[\theta](\frac{y}{\lambda_{t-1}} - \theta)}{\| \frac{y}{\lambda_t} - \theta - \alpha[\theta](\frac{y}{\lambda_{t-1}} - \theta) \|} \right\|_2,$$

$$\leqslant r_{\lambda_{t-1}} |\alpha[\theta] - 1| + \Big\| \frac{y}{\lambda_t} - \theta - \alpha[\theta].(\frac{y}{\lambda_{t-1}} - \theta) \Big\|. \tag{20}$$

Hence, after some simplifications:

$$\hat{\theta}^{(\lambda_t)} \in \mathcal{B}\left(\theta, r_{\lambda_{t-1}}(1 + |1 - \alpha[\theta]|) + \left\|\frac{y}{\lambda_t} - \theta - \alpha[\theta](\frac{y}{\lambda_{t-1}} - \theta)\right\|_2\right). \qquad \square$$

**Remark 8.** In the case that $\|y/\lambda_{t-1}\| \leqslant \|y/\lambda_{t-1} - \theta\| \leqslant 1$ then with the definition of $\alpha[\theta]$ and the Cauchy-Schwartz inequality one has that $1 + |\alpha[\theta] - 1| \leqslant \frac{\lambda_{t-1}}{\lambda_t}$. This means that the multiplicative ratio in front of $r_{\lambda_{t-1}}$ is $\lambda_{t-1}/\lambda_t$. In [11, Proposition 3], the bound obtained would only lead to the smaller ratio: $\sqrt{\lambda_{t-1}/\lambda_t}$.

**Remark 9.** From the proof of Theorem 7 in [21], it holds that for $\lambda < \lambda_{\max}$ then

$$\left\|\hat{\theta}^{(\lambda)}\right\| \leqslant \frac{\|y\|}{\lambda} \Leftrightarrow \hat{\theta}^{(\lambda)} \in B\left(0, \frac{\|y\|}{\lambda}\right). \tag{21}$$

## C.2 The complete screening rule (EDDP+)

Let us now consider the EDDP+ screening rule [21] relying on the property (15): $\hat{\theta}^{(\lambda_t)} \in \mathcal{B}(\hat{\theta}^{(\lambda_{t-1})} + \frac{1}{2}v^\perp(\lambda_{t-1}, \lambda_t), \frac{1}{2}\|v^\perp(\lambda_{t-1}, \lambda_t)\|_2)$. Using the same technique as for Proposition 4, we can strengthen our previous proposition with the following result.

**Proposition 5.** *Assume that $\lambda_{t-1} < \lambda_{\max}$, $\theta \in \Delta_X$ is a dual feasible point and $r_{\lambda_{t-1}} > 0$ is a radius satisfying $\hat{\theta}^{(\lambda_{t-1})} \in \mathcal{B}(\theta, r_{\lambda_{t-1}})$. Define $\alpha[\theta]$ as in (19),*

$$r_{\lambda_t} = \frac{|1 - \alpha[\theta]| + 1 + \alpha[\theta]}{2}r_{\lambda_{t-1}} + \frac{1}{2}\left\|\frac{y}{\lambda_t} - \theta - \alpha[\theta](\frac{y}{\lambda_{t-1}} - \theta)\right\|_2$$

$$+ \frac{\left\|\frac{y}{\lambda_t} - \frac{y}{\lambda_{t-1}}\right\|_2 r_{\lambda_{t-1}}}{2\|\frac{y}{\lambda_{t-1}} - \theta\|_2^2}\left(3\left\|\frac{y}{\lambda_{t-1}} - \theta\right\|_2 + 2r_{\lambda_{t-1}}\right)$$

*and*

$$v^\perp(\theta, \lambda_{t-1}, \lambda_t) = \frac{y}{\lambda_t} - \theta - \alpha[\theta](\frac{y}{\lambda_{t-1}} - \theta). \tag{22}$$

*Then $\hat{\theta}^{(\lambda_t)} \in \mathcal{B}\left(\theta + \frac{1}{2}v^\perp(\theta, \lambda_{t-1}, \lambda_t), r_{\lambda_t}\right)$.*

*Proof.* As before, we do not know exactly $\hat{\theta}^{(\lambda_{t-1})}$ but we know that denoting

$$v^\perp(\theta', \lambda_{t-1}, \lambda_t) = \frac{y}{\lambda_t} - \theta' - \alpha[\theta'](\frac{y}{\lambda_{t-1}} - \theta') \tag{23}$$

with

$$\alpha[\theta'] = \left(\frac{\langle\frac{y}{\lambda_{t-1}} - \theta', \frac{y}{\lambda_t} - \theta'\rangle}{\|\frac{y}{\lambda_{t-1}} - \theta'\|_2^2}\right)_+, \tag{24}$$

we have

$$\hat{\theta}^{(\lambda_t)} \in \bigcup_{\theta' \in \mathcal{B}(\theta, r_{\lambda_{t-1}})} \mathcal{B}\left(\theta' + \frac{1}{2}v^\perp(\theta', \lambda_{t-1}, \lambda_t), \frac{1}{2}\|v^\perp(\theta', \lambda_{t-1}, \lambda_t)\|_2\right).$$

Our goal is to find a ball centered at $\theta + \frac{1}{2}v^\perp(\theta, \lambda_{t-1}, \lambda_t)$ that contains all these balls, thus containing $\hat{\theta}^{(\lambda_t)}$. First, reminding (20)

$$\left\|v^\perp(\theta', \lambda_{t-1}, \lambda_t)\right\|_2 = \min_{\alpha \in \mathbb{R}_+} \left\|\frac{y}{\lambda_t} - \theta' - \alpha(\frac{y}{\lambda_{t-1}} - \theta')\right\|_2$$

$$\leqslant \max_{\theta' \in \mathcal{B}(\theta, r_{\lambda_{t-1}})} \min_{\alpha \in \mathbb{R}_+} \left\|\frac{y}{\lambda_t} - \theta' - \alpha(\frac{y}{\lambda_{t-1}} - \theta')\right\|_2$$

$$\leqslant r_{\lambda_{t-1}}|1 - \alpha[\theta]| + \left\|\frac{y}{\lambda_t} - \theta - \alpha[\theta](\frac{y}{\lambda_{t-1}} - \theta)\right\|_2.$$

We continue as

$$\theta' + \frac{1}{2}v^\perp(\theta',\lambda_{t-1},\lambda_t) - \theta - \frac{1}{2}v^\perp(\theta,\lambda_{t-1},\lambda_t)$$

$$= (\theta'-\theta) + \frac{1}{2}\Big(\frac{y}{\lambda_t} - \theta' - \alpha[\theta'](\frac{y}{\lambda_{t-1}} - \theta') - \frac{y}{\lambda_t} + \theta + \alpha[\theta](\frac{y}{\lambda_{t-1}} - \theta)\Big)$$

$$= \frac{1}{2}\Big(\theta'-\theta - (\alpha[\theta']-\alpha[\theta])(\frac{y}{\lambda_{t-1}} - \theta') + \alpha[\theta](\theta'-\theta)\Big).$$

Taking the norm on both sides of the previous display,

$$\left\|\theta' + \frac{1}{2}v^\perp(\theta',\lambda_{t-1},\lambda_t) - \theta - \frac{1}{2}v^\perp(\theta,\lambda_{t-1},\lambda_t)\right\|_2 \leqslant \frac{1+\alpha[\theta]}{2}\left\|\theta'-\theta\right\|_2 + \frac{|\alpha[\theta']-\alpha[\theta]|}{2}\left\|\frac{y}{\lambda_{t-1}} - \theta'\right\|_2.$$

Now, reminding that $x \mapsto (x)_+$ is a 1-Lipschitz function,

$$|\alpha[\theta']-\alpha[\theta]| \leqslant \left| \frac{\langle \frac{y}{\lambda_{t-1}} - \theta', \frac{y}{\lambda_t} - \theta'\rangle}{\|\frac{y}{\lambda_{t-1}} - \theta'\|_2^2} - \frac{\langle \frac{y}{\lambda_{t-1}} - \theta, \frac{y}{\lambda_t} - \theta\rangle}{\|\frac{y}{\lambda_{t-1}} - \theta\|_2^2} \right|$$

$$= \left| \frac{\langle \frac{y}{\lambda_{t-1}} - \theta', \frac{y}{\lambda_t} - \frac{y}{\lambda_{t-1}}\rangle}{\|\frac{y}{\lambda_{t-1}} - \theta'\|_2^2} + 1 - \frac{\langle \frac{y}{\lambda_{t-1}} - \theta, \frac{y}{\lambda_t} - \frac{y}{\lambda_{t-1}}\rangle}{\|\frac{y}{\lambda_{t-1}} - \theta\|_2^2} - 1 \right|$$

$$= \left| \frac{\langle \|\frac{y}{\lambda_{t-1}} - \theta\|_2^2(\frac{y}{\lambda_{t-1}} - \theta') - \|\frac{y}{\lambda_{t-1}} - \theta'\|_2^2(\frac{y}{\lambda_{t-1}} - \theta), \frac{y}{\lambda_t} - \frac{y}{\lambda_{t-1}}\rangle}{\|\frac{y}{\lambda_{t-1}} - \theta'\|_2^2\|\frac{y}{\lambda_{t-1}} - \theta\|_2^2} \right|$$

$$\leqslant \frac{\left\|\frac{y}{\lambda_t} - \frac{y}{\lambda_{t-1}}\right\|_2}{\|\frac{y}{\lambda_{t-1}} - \theta'\|_2^2\|\frac{y}{\lambda_{t-1}} - \theta\|_2^2}\Big( \|\frac{y}{\lambda_{t-1}} - \theta'\|_2 \left|\|\frac{y}{\lambda_{t-1}} - \theta\|_2^2 - \|\frac{y}{\lambda_{t-1}} - \theta'\|_2^2\right| + \|\theta - \theta'\|_2 \|\frac{y}{\lambda_{t-1}} - \theta'\|_2^2\Big)$$

$$\leqslant \frac{\left\|\frac{y}{\lambda_t} - \frac{y}{\lambda_{t-1}}\right\|_2}{\|\frac{y}{\lambda_{t-1}} - \theta'\|_2\|\frac{y}{\lambda_{t-1}} - \theta\|_2^2}\Big( 2\|\frac{y}{\lambda_{t-1}} - \frac{\theta'+\theta}{2}\|_2\|\theta - \theta'\|_2 + \|\theta - \theta'\|_2 \|\frac{y}{\lambda_{t-1}} - \theta'\|_2\Big)$$

$$\leqslant \frac{\left\|\frac{y}{\lambda_t} - \frac{y}{\lambda_{t-1}}\right\|_2 \|\theta - \theta'\|_2}{\|\frac{y}{\lambda_{t-1}} - \theta'\|_2\|\frac{y}{\lambda_{t-1}} - \theta\|_2^2}\Big( 2\|\frac{y}{\lambda_{t-1}} - \theta\|_2 + \|\theta - \theta'\|_2 + \|\frac{y}{\lambda_{t-1}} - \theta\|_2 + \|\theta - \theta'\|_2\Big).$$

$$(25)$$

where the second inequality comes from the triangle inequality and the Cauchy-Schwartz Inequality, and the third is obtained by factorizing the difference of squares. Plugging this in the former, we get:

$$\left\|\theta' + \frac{1}{2}v^\perp(\theta',\lambda_{t-1},\lambda_t) - \theta - \frac{1}{2}v^\perp(\theta,\lambda_{t-1},\lambda_t)\right\|_2$$

$$\leqslant \frac{1+\alpha[\theta]}{2}\left\|\theta'-\theta\right\|_2 + \frac{1}{2}\frac{\left\|\frac{y}{\lambda_t} - \frac{y}{\lambda_{t-1}}\right\|_2 \|\theta - \theta'\|_2}{\|\frac{y}{\lambda_{t-1}} - \theta\|_2^2}\Big( 3\left\|\frac{y}{\lambda_{t-1}} - \theta\right\|_2 + 2\left\|\theta - \theta'\right\|_2\Big).$$

One could check that there exists $\theta' \in \mathcal{B}(\theta, r_{\lambda_{t-1}})$ satisfying $\hat{\theta}^{(\lambda_t)} \in \mathcal{B}\big(\theta' + \frac{1}{2}v^\perp(\theta',\lambda_{t-1},\lambda_t), \frac{1}{2}\left\|v^\perp(\theta',\lambda_{t-1},\lambda_t)\right\|_2\big)$ and so combining the last inequality with (25)

$$\left\|\hat{\theta}^{(\lambda_t)} - \theta - \frac{1}{2}v^\perp(\theta,\lambda_{t-1},\lambda_t)\right\|_2 \leqslant \left\|\hat{\theta}^{(\lambda_t)} - \theta' - \frac{1}{2}v^\perp(\theta',\lambda_{t-1},\lambda_t)\right\|_2$$

$$+ \left\|\theta' + \frac{1}{2}v^\perp(\theta',\lambda_{t-1},\lambda_t) - \theta - \frac{1}{2}v^\perp(\theta,\lambda_{t-1},\lambda_t)\right\|_2$$

$$\leqslant \frac{|1-\alpha[\theta]|+1+\alpha[\theta]}{2}r_{\lambda_{t-1}} + \frac{1}{2}\left\|\frac{y}{\lambda_t} - \theta - \alpha[\theta](\frac{y}{\lambda_{t-1}} - \theta)\right\|_2$$

$$+ \frac{\left\|\frac{y}{\lambda_t} - \frac{y}{\lambda_{t-1}}\right\|_2 r_{\lambda_{t-1}}}{2\|\frac{y}{\lambda_{t-1}} - \theta\|_2^2}\Big( 3\left\|\frac{y}{\lambda_{t-1}} - \theta\right\|_2 + 2r_{\lambda_{t-1}}\Big) \qquad \square$$