[Reviews · NeurIPS 2015]

Submitted by Assigned_Reviewer_1

The paper proposes safe screening rules for sparse multitask and multiclass learning models. The rules are obtained by extending GAP safe rules proposed in [10]. The applicability of these rules in several specific problem instances is discussed. The paper also shows that EDPP rules [20] can be unsafe and proposes modifications to make them safe.

Although the novelty is still limited, in the sense that the main screening technique is borrowed from [10], I feel the overall contribution, including the fix of EDPP rules to make these safe, is sufficient for publication.

Paper has some typos: line 89: promoting a few line.. line 97: missing ref for group lasso line 219: safe screening rules "are" the most.. line 221: For to hold.. line 313: missing ref for Liblinear

Summary: The paper extends GAP safe rules [10] to sparse multitask and multiclass problems. It is a well written paper (except a few typos) and makes a clear contribution. It also shows that EDPP rules [20] can be unsafe and proposes modifications to make them safe.

Submitted by Assigned_Reviewer_2

Summary:

This paper presents an unprecedentedly fast method for eliminating variables during iterations of multi-task group lasso that is provably safe, meaning that all variables that are eliminated would ultimately obtain zero weights when running vanilla group lasso.

This method is iterative; as the primal solver converges, it eliminates an increasing number of variables.

The authors compare their method to previous methods and demonstrate that all previous methods are either unsafe or are substantially slower for small duality gaps.

The authors describe how their method should be applied to specific cases of group lasso, including l1 and l1/l2 regularized logistic regression, and present real applications where their method achieves a substantial speed-up over vanilla group lasso and an existing method for small duality gap thresholds.

General suggestions: 1. The primary situation where this method obtains substantial speed improvements over alternatives is where the duality gap threshold is extremely small.

To make the importance of the work more convincing, the authors need to demonstrate that this is a common situation or at least a situation that occurs in some important group lasso settings.

If the duality gap must be small in one of the real examples the author use in Section 4, then stating this and providing a citation for it would be sufficient. 2. Some of the derivations skipped many steps.

The paper would be much easier to follow if these derivations were fleshed out in more detail, either in the main text or in the Supplement.

I have noted the places where this was the case in "Specific comments and suggestions." 3. There are many grammar and word choice errors, including in the statements of many of the propositions.

To make the paper sufficiently clear for readers to grasp what the authors have accomplished, the authors need to fix these errors.

I have pointed out the most problematic ones in "Specific comments and suggestions."

Specific comments and suggestions, main text: 1. In the Introduction, the authors discuss the lasso problem.

In Section 2, they discuss the group lasso problem.

They should explicitly state that the lasso is a specific case of the group lasso and maybe re-word the introduction to discuss the group lasso problem. 2. There is a missing citation in Remark 1 in Section 2.1. 3. The authors should cite Fermat's condition at the beginning of Section 2.2. 4. The authors should either derive the dual formulation of the group lasso that they present in Section 2.2 in the Supplement or cite the paper in which it was derived. 5. In the first "if and only if" statement in Section 2.3, I am convinced that 0 being a primal solution implies the next statement, but it is not obvious why the converse is true.

The authors should justify the converse in more detail. 6. The authors should add a conclusion sentence to section 2.3 that summarizes what they have shown. 7. In the beginnings of the subsections of Section 2.6, the word "Remind" should be replaced with "Remember." 8. It is not obvious why the sentence "Since theta^(gamma) maximizes D_gamma..." near the beginning of Section 2.6 is true, so the authors should justify this in greater detail. 9. Remark 4, which refers to equation (15), should be moved after equation (15).

It is also not obvious why the first sentence is true, so the authors should justify it in greater detail. 10. When summarizing the Dynamic GAP Safe rule in Section 2.6, the authors state that dynamic safe screening rules are "the most efficient."

They either need to rigorously prove that making more efficient rules would be impossible or replace the phrase "the most efficient" with something like "more efficient than existing methods in practice." 11. The way that Proposition 1 in Section 2.6 is worded is very confusing, so the authors should re-word it. 12. In Remark 7, the authors say that the active set obtained by GAP safe rules allow the solver to "identify relevant features earlier."

They should say what method they are comparing to.

Do they mean "earlier than previous methods would?" 13. In Section 3, the authors frequently use a variable z.

They should define z at the beginning of the section. 14. The authors should add conclusion sentences to Sections 3.1 and 3.2 that summarize what they have shown. 15. I am not sure if the formulation of the logistic regression in (12) in Section 3.3 is correct.

I think that the y_i should be in front of the parentheses and that there should be a term for the other class. 16. I am not sure if the formulation of the multinomial logistic regression in (14) is correct.

I think that the Y_i should be in front of the second summation. 17. It is not obvious why the final sentence of Section 3.3 is true, so the authors should justify it in greater detail. 18. It is not obvious why the final sentence of Section 3.4 before Remark 9 is true, so the authors should justify it in greater detail. 19. Table 1 in Section 3 should be referred to somewhere in the main text. 20. In Section 4, the authors describe how they implemented the GAP Safe rule.

They should make the code publicly available so that others can easily use their algorithm when solving group lasso problems. 21. In each of the applications the authors describe in Section 4, the author should explain why the optimization problem they are using is the right way to solve the problems or cite papers that have these explanations. 22. The heatmaps in Figures 1-3 are confusing.

They might be clearer if the axes went in the opposite directions.

The authors should also explicitly label the color bars in the figures. 23. In Section 4 and elsewhere, the authors sometimes call their method "GAP SAFE (Sphere)" and sometimes call it "GAP Safe (dynamic)."

The authors should either always use the same terminology or, if these methods are not the same, explicitly define them and state the difference. 24. In Section 4, the authors compare to different existing methods for different datasets.

The authors should explain why they do not compare to every method for every dataset. 25. In Section 4.2, the authors should summarize the results presented in Figure 2 in the text. 26. In Section 4.2, the authors should describe the "safe rule close in spirit to the Slores itself" or refer to the description in the Supplement. 27. The sentence in Section 5 starting with "Extensions to other type" is confusing, so the authors should re-word it.

Specific comments and suggestions, supplement: 1. The phrasing of Proposition 2 in Section A.1 is a little confusing, so the authors should re-word it. 2. In the second paragraph in Section B, the authors should replace "sure" with "safe." 3. It is not obvious why the last sentence of Section B is true, so the authors should explain it in greater detail. 4. The phrasing of Proposition 4 in Section C.1 is a little confusing, so the authors should re-word it. 5. In Section C.1, it is not obvious why the choice for alpha[theta] is optimal when r_gamma_t-1 = 0, so the authors should explain why this is true in more detail. 6. The phrasing of Proposition 5 in Section C.2 is a little confusing, so the authors should re-word it. 7. When deriving the bound on |alpha[theta']- alpha[theta]| in Section C.2, it is not obvious why the second and third inequality are true, so the authors should provide additional steps. 8. In section C.2, it was not obvious why the final inequality is true, so the authors should provide additional steps.
Summary: This paper presents an unprecedentedly fast method for eliminating variables during iterations of multi-task group lasso that is provably safe, meaning that all variables that are eliminated would ultimately obtain zero weights when running vanilla group lasso.

However, this paper needs to clarify the importance of the situations where it provides a substantial speed-up, some of the derivations that it presents, and the wording in many of the propositions.

Submitted by Assigned_Reviewer_3

This paper addresses a new safe dynamic rule for l1 and group-l1 regularized problems, which allows us to screen out irrelevant variables during optimization. This helps us to speed-up the optimization procedure which tends to be expensive when the data dimensionality is high. A notable strength of the proposed method is its wide applicability: it only requires a primal converging algorithm (+ strong convexity of the dual objective and weak duality). The authors proved that the procedure is safe, i.e. it does not screen out relevant variables. The authors also presented some numerical results showing the efficacy of the proposed method.

I think the paper is well-written and the technical detail is clearly stated. The wide applicability of the method would be helpful for machine learning users in practice. However, most technical parts were already established in the ICML 2015 paper (O. Fercoq et al.). The main contribution of this paper is to extend the scope of the ICML paper from lasso specific to generalized linear models. In particular, Eq. (11) enabled the easy compuation of the dual feasible point for many loss functions other than lasso. Proposition 1 guaranteed that the procedure is still safe under Eq.(11). Other than Eq.(11) and Proposition 1, I think the results in the paper are already established in the ICML paper. Hence I think the technical progress made in this paper is relatively minor.

[minor comment] Some references are missing in the paper as [?].
Summary: This paper extends the safe dynamic rule for lasso presented in ICML 2015 to generalized linear models. As most technical parts were already established in the ICML paper, I think the technical progress made in this paper is relatively minor and it is not strong enough to be accepted.

Submitted by Assigned_Reviewer_4

An interesting safe screening method for learning model with sparse regulazer. The key is to dynamically construct a safe region to bound the dual constraint for identifying inactive features. In lines 51-52, the authors mentioned sequential safe rules might wrongly discard relevant features. This is not true. Sequential safe rules are guaranteed to be safe. See the proof provided in one relevant paper that is provided in [1]. By properly implementing the algorithm, not relevant feature will be discarded.

Also by sequentially decreasing the value of regularization parameter in a proper way, the sequential solver might converge faster. [1] also provided a good way to determine the step for decreasing the value of regularization parameter. The result presented in Figure-2, right chart might be misleading. It is obvious that the bound provided by the proposed method in its sequential version is looser than the one provided in [1].

The paper needs to address the above concerns.

[1] Safe Screening with Variational Inequalities and Its Application to Lasso. The 31st International Conference on Machine Learning (ICML), 2014.
Summary: An interesting safe screening method for learning model with sparse regulazer. The key is to dynamically construct a safe region to bound the dual constraint for identifying inactive features.

Submitted by Assigned_Reviewer_5

The work extends previous ideas of dynamic safe rule computations. The novelty mostly lies in the extension of previous ideas to a more general framework with applications to numerous learning problems. The paper is mostly theoretical and generally sound.

The paper is generally well written. There are references missing at places, and some parts of the text that are a bit confusing (like the end of Section 2.4). The results are minimally described, and not always clear enough to really appreciate the benefit of the proposed method. Also, for the sake of completeness, the authors should explain the duality gap, that is an important concept, . Finally, the online or dynamic aspect of the method is not very clearly explained. Maybe a general figure would be of great help to clearly present the main steps of the framework.
Summary: The papers presents a technique to speed-up solver in learning algorithms, by identifying safe regions that could be excluded from the computation and lower the overall complexity. It extends previous works and generalises the dynamic safe regions computation to a large class of problems.

Author Feedback
Author rebuttal: First, we would like to thank the reviewers for their valuable comments,
and especially Assigned_Reviewer_2 whose remarks will greatly improve the reading.

Apart from minor remarks that will be addressed upon acceptance, we would like to raise the following points:

2) We were surprised that Assigned_Reviewer_6 disagree that EDDP and its variants are unsafe. Let us make our point clearer here: as already exposed in Fercoq et al. (2015), any sequential rule that relies on the EXACT knowledge of the dual solution is doomed to be unsafe (except for the case where the dual point is simply $y/\lambda$). Indeed, so far sequential rules have all been built on properties of dual optimal points that might fail to be true for their approximation. Since in practice dual solutions are approximated by a numerical solver, it might happen that relevant variables are wrongly disregarded. This could prevent the solver from converging without further checks, both for the primal and dual solutions (an illustration is provided in the supplementary material). Note that this drawback is shared by all variants of EDDP among others.

3) Our contribution is judged incremental by Assigned_Reviewer_3, so we want to emphasize the following points:

-we have adapted dynamic safe screening rules to a wide variety of problems

(various loss and penalty terms while Fercoq et al. (2015) focused only on the Lasso ie square l2 loss and l1 regularization). Doing so, we have unified a large collection of recent papers on safe screening rules such as [4,5,8,10,20,21,22].

-we have simplified and generalized the proof that Gap Safe rules are safe and converging. Note, for instance, that the proof from Fercoq et al. (2015) does not apply in our framework, since it heavily relies on square l2 loss properties and Euclidean geometry.

-we have proposed a correction to make standard sequential safe rules really safe, leveraging the duality gap information (see discussion above).

Also note that many papers on safe rules, relying on adaptation of original rules to more evolved models, have recently been accepted in top ML conferences/journals, e.g.:

"Safe Screening for Multi-Task Feature Learning with Multiple Data Matrices", ICML 2015

"Detecting genetic risk factors for Alzheimer's disease in whole genome sequence data via Lasso screening". ISBI, 2015

"Lasso screening rules via dual polytope projection". JMLR, 2015.

"A safe screening rule for sparse logistic regression". NIPS, 2014

"Safe screening with variational inequalities and its application to lasso". ICML, 2014

"Lasso screening rules via dual polytope projection". NIPS, 2013.

Our proposed contribution can encompass these papers and correct the common aforementioned limits they suffer from.

4) "It is obvious that the bound provided by the proposed method in its sequential version is looser than the one provided in [1]"(Assigned_Reviewer_6)

This is indeed true, since we have to account for the limited approximation of the dual solution we obtained for the previous $\lambda$. This is the price to pay for not accounting for dual solutions inaccuracies .

5)"by sequentially decreasing the value of the regularization parameter in a proper way, the sequential solver might converge faster" (Assigned_Reviewer_6)

GAP Safe strategies also benefits from this warm start step.